# Unveiling Fabrication and Environmental Remediation of MXene-Based Nanoarchitectures in Toxic Metals Removal from Wastewater: Strategy and Mechanism

**DOI:** 10.3390/nano10050885

**Published:** 2020-05-04

**Authors:** Yassmin Ibrahim, Amal Kassab, Kamel Eid, Aboubakr M. Abdullah, Kenneth I. Ozoemena, Ahmed Elzatahry

**Affiliations:** 1Center for Advanced Materials, Qatar University, Doha 2713, Qatar; yi1511403@student.qu.edu.qa (Y.I.); kamelame@outlook.com (K.E.); 2Biomedical and Biological Engineering Department, McGill University, Montreal, QC H3A 0G4, Canada; amal_kassab@hotmail.com; 3Molecular Sciences Institute, School of Chemistry, University of the Witwatersrand, Private Bag 3, P O Wits, Johannesburg 2050, South Africa; kenneth.ozoemena@wits.ac.za; 4Materials Science and Technology Program, College of Arts and Sciences, Qatar University, PO Box 2713, Doha 2713, Qatar

**Keywords:** MXene, wastewater treatment, toxic metal removal, Ti_3_C_2_ nanocomposites, heavy metals

## Abstract

Efficient approaches for toxic metal removal from wastewater have had transformative impacts to mitigating freshwater scarcity. Adsorption is among the most promising purification techniques due to its simplicity, low cost, and high removal efficiency at ambient conditions. MXene-based nanoarchitectures emerged as promising adsorbents in a plethora of toxic metal removal applications. This was due to the unique hydrophilicity, high surface area, activated metallic hydroxide sites, electron-richness, and massive adsorption capacity of MXene. Given the continual progress in the rational design of MXene nanostructures for water treatment, timely updates on this field are required that deeply emphasize toxic metal removal, including fabrication routes and characterization strategies of the merits, advantages, and limitations of MXenes for the adsorption of toxic metals (i.e., Pb, Cu, Zn, and Cr). This is in addition to the fundamentals and the adsorption mechanism tailored by the shape and composition of MXene based on some representative paradigms. Finally, the limitations of MXenes and their potential future research perspectives for wastewater treatment are also discussed. This review may trigger scientists to develop novel MXene-based nanoarchitectures with well-defined shapes, compositions, and physiochemical merits for efficient, practical removal of toxic metals from wastewater.

## 1. Introduction

The scarcity of freshwater resources is the most perilous issue threatening humanity due to only 2.5% of the water on the earth being fresh and accessible, while the remaining amount is saline (97%) and inaccessible [1]. The industrial and agricultural activities waste nearly 80% of their freshwater, and only 20% of this used water is treated and reused [1]. The manufacturing, electrolysis, metal electroplating, metallurgy, mineral extraction, and ceramics industries produce huge amounts of wastewater containing As, Zn, Cu, Pb, Hg, Cr, Ni, and Cd, which are highly poisonous, carcinogenic, and non-degradable [1,2,3]. After irrigation, these metals can easily leak into freshwater through the soil to be absorbed by aquatic plants, animals, and finally, by humans through the food chain, resulting in serious health problems [4]. Various efforts have been dedicated to removing toxic metals from wastewater using different physical, chemical, and biological methods, which vary in performance, environmental impact, and associated limitations [4,5,6]. For instance, chemical precipitation depends on using a massive amount of chemicals (i.e., lime and limestone) generating secondary pollution. Membrane-based technology (i.e., ultrafiltration, reverse osmosis, and electrodialysis) is highly costly and requires special devices, whereas biological methods (i.e., bacteria, fungi, and microorganism) are impractical [4,5,6,7].

Unlike these treatment methods, adsorption is a facile, efficient, eco-friendly, and cost-effective strategy that does not require capital investment; however, both regeneration and selectivity remain challenges [4]. Massive adsorbents are generally used, such as metals oxides (i.e., Fe_2_O_3_ and TiO_2_), carbon (i.e., carbon nanotubes, graphene, and activated carbon), and polymers, which differ in their adsorption capability and cost as well as preparation difficulties [4,5,6,7,8].

MXene’s new family of carbonitrides, nitrides, and metal carbide, developed in collaboration with Barsoum’s and Yury’s groups, emerged as highly promising adsorbents in environmental remediation applications [9,10]. This is due to the unique physicochemical merits of MXene, such as hydrophilicity, large surface area, activated metallic hydroxide sites, accessible adsorption sites, electron richness, and massive adsorption capacity. The interlayer spacing between MXene nanosheets (<2 Å) provides strong trapping sites for cations of radii <4.5 Å [9]. MXene can be easily prepared from various inexpensive, safe, and abundant products. MXene is denoted by the general formula M_n+1_X_n_T_x_, where (n = 1–3), M is an early transition metal (e.g., Ti, V, Cr, and Nb), X is carbon and/or nitrogen, A is an element from group 13 or 14 of the periodic table (Al, Si, Sn, In, etc.), and T_x_ is the surface terminal (e.g., OH, O, Cl, or F) [9,11]. Substantial progress has occurred in the rational design of around 30 types of MXenes-based nanostructures, such as Ti_3_C_2_T_x_, Ti_2_CT_x_, Nb_4_C_3_T_x_, Ta_4_C_3_T_x_, V_2_CT_x_, and (V_0.5_, Cr_0.5_)_3_C_2_); however, Ti_3_C_2_T_x_ is most commonly used for wastewater treatment applications [9,11,12]. Several groups published reviews in the fields of the synthesis and applications of MXene for environmental remediation, but educational reviews related to the use of MXene for toxic metal removal are lacking [9,11,12,13,14,15]. Due to the explosion of publications and the continual development of the preparation of MXenes for water treatment applications, timely updates on this field are required.

In pursuit of this aim, this review is dedicated to highlighting the salient features related to controlling the shape and composition of MXenes’ nanostructures for toxic metal removal from wastewater, including fabrication strategies and characterization techniques with their formation mechanisms. The advantages and limitations of MXenes as adsorbents of toxic metals (i.e., Pb, Cu, Zn, and Cr) are reviewed, along with their fundamentals and adsorption mechanism, supported by various representative paradigms. Then, outlooks and perspectives for future research in the field of toxic metal are suggested.

## 2. Methods for Toxic Metal Removal from Wastewater

Several wastewater management methods have been developed over the past few decades [16]. Traditional toxic metal removal methods include precipitation, ion exchange, reverse osmosis, electrokinetic methods, membrane filtration, and adsorption [17]. Although some of these methods are often used interchangeably, such as ion exchange and adsorption, most are used in combination to isolate the toxic metal and other particles using reverse osmosis or isolation chambers. Then, toxic metal solutes are transformed into solid compounds either using a chelating agent, precipitation, coagulation, adsorption, or electrokinetics; then these newly formed particles are filtered using various membranes or other collection methods. Preprocessing steps might include the addition of lime or other agents to obtain an optimal pH level based on the type of method used, as well as adjustments to temperature and fluid flow. Post-operational procedures vary but primarily include isolation of solid precipitate, dewatering, transferring the solid waste to a landfill, and further treatment of the remaining liquid until an acceptable set of specifications is reached that allows it to be returned to nature [18,19,20].

Obvious drawbacks in traditional wastewater treatment range from the creation of secondary pollution in the case of chemical precipitation, to susceptibility and fouling by organic matter in the case of membranes, a limited ion removal margin, and a high energy cost in the case of electrokinetic remediation. Of all developed methods, adsorption has shown promise due to its simplicity and speed of operation, especially with the emergence of adsorbents that circumvent the need for secondary processing and pollution. Evermore organic and inorganic materials are being introduced as possible adsorption agents for toxic metal removal, such as rice husk, palm fiber, eggshells, chitosan, clinoptilolite, various bacterial species, plants, clay, activated carbon, and various types of nanomaterials [21,22,23,24].

Of all available adsorbents, nanoparticles, whether organic or inorganic, have the most impressive adsorption abilities; some of them are already being introduced into the wastewater management industry. Nanomaterials as adsorbents are characterized by unique properties and high surface areas, which allow them to handle larger toxic metal concentrations in comparison with other chelating agents or adsorbents [24]. Their surface tunability using various activation methods may offer toxic metal selectivity, which in turn provides future recycling opportunities using these particles instead of disposing of them as toxic waste. Such nanomaterials include zeolites, bacterial exopolysaccharide, metal-organic frameworks (MOFs), algae, ceramics, and carbons, of which the most widely used nanoparticle in wastewater management is activated carbon, despite being a relatively expensive option [25,26,27,28,29,30].

Within this ever-growing category, upcoming 2D nanostructures are showing promise because they have higher adsorption abilities due to their large functionalized surfaces with unique features that allow them to be integrated into several wastewater treatment processes, either as adsorbents, or catalytic and/or antibacterial agents, in addition to functional membranes. The technology is still under development, and there are several apparent setbacks to the immediate commercialization of these products, including their rather short shelf lives and inability to be integrated within the existing framework of wastewater management [31].

## 3. Characterization of MXenes and Their Preparation

MXenes, as an upcoming group of promising 2D materials, have been the topic of various recent review articles and book chapters that have covered their unique properties, possible areas of application, and syntheses [12,32,33,34,35,36,37,38]. Therefore, this review only focuses on the characterization of specific MXene structures as adsorbents in wastewater management to highlight their particle entrapment methods and the role of particular surface functionalization in their operational efficiency.

As a term, “MXenes” is used to describe a range of nanoparticles that share the general formula M_n + 1_X_n_T_x_ (n = 1–3), where M can be any early transition metal—Sc, Ti, Zr, Hf, V, Nb, Ta, Cr, Mo, etc.; X is carbon or nitrogen; and T_x_ is surface termination that varies based on the method of fabrication and the properties required in the type of MXene formed. Therefore, these terminations can include but are not limited to oxygen, hydroxyl, or fluorine, which are the most common terminations. Research into MXenes using different raw materials and fabrication methods is still ongoing, including the emergence of new 3D MXene structures [38,39].

What makes MXenes particularly interesting is its layered structure of M (n + 1 layers) and X (n layers), which are considered 3D pores within their nanoscale structure [10]. Whereas most MXenes are composed of one transition metal each, some variations exist wherein more than one M element is present, either within a random distribution or in an ordered layered fashion. In general, synthesis of MXenes relies on the selective etching of A group elements from their MAX phase precursors. MAX phase compounds represent a large family of layered carbides and nitrides (X elements) sandwiched between M layers of early transition metal and A layers of mostly IIIA and IVA or 13 and 14 group metals such as Al or Si, but can also include P, S, Ga, Ge, As, Cd, In, Sn, Tl, and Pb [10]. The most common and commercially available MAX phase compound is Ti_3_AlC_2_; therefore, most MXenes studies are those with the formula Ti_3_C_2_T_x_, where T represents surface terminations such as −F, −OH, or −O. The preparation of Ti_3_C_2_T_x_ is based on the selective chemical etching method using hydrogen fluoride (HF) solution, followed by delamination using dimethyl sulfoxide (DMSO) under sonication. In addition to wet etching methods, dry etching is widely used to create surface terminations that are fluoride-free to enhance the surface activation of MXenes and increase suitability for some applications. Figure 1 illustrates the effects of various etching and delamination methods on created MXene formulae and the stacking of MXene layers [36].

The effects of various MXene preparation methods on their morphologies, structure, and surface terminations, are discussed with regard to the toxic metal in the following sections. Before detailing the efficiency of MXenes in removing toxic metal particles, one of the main obstacles in MXenes’ commercialization is its thermodynamically metastable state. MXenes have remarkable surface energy, which makes them particularly vulnerable to degradation. In particular, MXene solutions tend to oxidize within several days, forming titanium dioxide crystals at the edges of MXene flakes until the whole structure transforms into TiO_2_ and carbon sheets. Pristine and oxidized MXenes can be distinguished by observing the color difference, as pristine MXene flakes are greenish-black that turns into gray or cloudy white upon oxidization. MXene powders are more resilient in general and remain stable in an oxygen-containing atmosphere at temperatures below 200 °C [40]. For toxic metal remediation, TiO_2_ is considered an efficient adsorbent for various metals, such as mercury [41], copper [42], and chromium [43]. Therefore, several MXene compounds are intentionally aged under hydrothermal conditions to obtain TiO_2_ surface elements, which are discussed in more detail below.

Various stabilizing methods have been studied for MXene flakes using carbon nanoplates, by creating new MXene hybrids that are reinforced with carbon and MoS_2_, and using high energy mechanical milling in DMSO, demonstrating that high stability can be achieved by the addition of fluorine to their surfaces [44,45,46]. Future MXene compounds with greater stability and longer shelf lives are expected for wastewater management wherein MXenes flakes are under aquatic conditions. The fragility and fast degradability of MXenes have to be considered when designing remediation systems based on a targeted toxic metal. Fast decomposition of MXenes into environmentally friendly components can be considered advantageous, as it may facilitate post-operational handling of the resulting slurry in wastewater management. Further studies into the applications of MXenes in water treatment and their degradation after binding with toxic metals or other toxic wastes are essential to define their scopes of operation and ultimate efficiency.

## 4. MXenes for Toxic Metal Removal

The adsorptive ability of MXenes for toxic metals is mainly due to the surface groups formed during the alkalization–intercalation process. Ti_3_C_2_ is converted into Ti_3_C_2_(OH)_2_, which can be I-Ti_3_C_2_(OH)_2_, II-Ti_3_C_2_(OH)_2_, or III-Ti_3_C_2_(OH)_2_ based on its location on the Ti atom. I-Ti_3_C_2_(OH)_2_ denotes those located on the top site of the inner Ti atom, the II-Ti_3_C_2_(OH)_2_ group lies above the top site of the C atom, and III-Ti_3_C_2_(OH)_2_ is a combination of the two, sharing some similarities with both types. These different surface groups have different energy levels, with I-Ti_3_C_2_(OH)_2_ being the highest in energy level and III-Ti_3_C_2_(OH)_2_ being the lowest.

Alkylated-MXenes are the ideal candidates for adsorption of various toxic metals, such as Pb, Cr, Zn, Pd, and Cd [47]. Mainly, cations can be intercalated in MXene layers due to the thin interlayer spacing of <2 Å, increasing their spacing radii to <4.5 Å. Therefore, unlike other 2D materials [31], MXenes can effectively trap metals within their interlayer spacing as an additional removal mechanism, increasing their potency.

The following represents the latest studies performed on toxic metal remediation using various MXene configurations. In most cases, the efficiency of MXenes in toxic metal adsorption is superior, and in some cases, it is comparable to those performed using different adsorbents. Given the surface tunability of MXenes, various studies integrated new surface groups, which proved to be effective in metal adsorption and enhanced MXenes efficiency and selectivity. The electrical conductive properties and hydrophilicity of MXene flakes allow them to be coupled with electrochemical separation and deionization methods used in wastewater management [47]. Table 1 summarizes the latest studies conducted on MXenes for toxic metal removal.

### 4.1. Sorption of Lead by MXenes

Lead is one of the most common pollutants and dangerous toxins (class 2 hazard) that is produced regularly as a by-product of various industrial and agricultural processes, such as metal plating, mining, batteries, fertilizers, and paint manufacturing [66]. High concentrations of lead affect the central nervous system in addition to the renal, cardiovascular, reproductive, hematopoietic, and gastrointestinal systems. Thus, the maximum allowable limit of lead discharge into wastewater was set to 0.10 mg/L by the World Health Organization (WHO) [67].

Adsorption, as a means of lead removal, has been considered using several nanomaterials; among them, glycol-graphene oxide exhibited an adsorption capacity of 146 mg/g of lead with a contact time of 48 h and pH levels ranging from 2 to 6 [67]. The 79.8 mg/g adsorption capacity of Pb was achieved using magnetic chitosan graphene oxide within only 0.67 h at pH 5 [68]. A graphene oxide compound with 2,2’-dipyridylamine increased the adsorption capacity of Ph(II) to 369.79 mg/g within 0.67 h at pH 7.4 under ultrasonic conditions [69]. In comparison with these findings, experimental studies of MXene adsorption of lead show promise, particularly if coupling the mechanisms of these compounds with other known adsorption-inducing factors, such as ultrasonic amplification or surface modifications, which might yield even more effective results.

Alkylated 2D MXene Ti_3_C_2_(OH/ONa)_x_F_2−x_ achieved a significant adsorption efficiency of Pb [48]. This work was based on earlier theories on the effects of surface groups on MXene adsorption [70]; thus, the compound was prepared by selective chemical exfoliation followed by alkalization intercalation. Preparation of these alk-MXene uses a simple etching method of commercial MAX flakes (Ti_3_AlC_2_) with an average size of 3 nm in an HF solution, resulting in 50–100 nm gaps between titanium sheets that contain OH and F groups. This was followed by an alkalization treatment with NaOH, causing the intercalation of Na ions within the structure and a reduction in F. These Ti_3_C_2_(OH/ONa)_x_F_2−x_ flakes maintain their hexagonal structure, forming multilayer stacks (~20) with a spacing of 1.5 nm along the c-axis. Figure 2a shows an SEM image of alk-MXene after exfoliation in HF, whereas Figure 2b shows the high-resolution transition electron microscope (HTEM) image of the resulting alk-MXene.

Sorption of lead ions Pb(II) using this compound was shown to be a function of pH with more preferential sorption at higher levels within the pH range of 5–7, indicating a mechanism of action that relies on ion exchange. The low affinity of MXenes toward Pb ions at low pH might allow for the regeneration of used flakes in acidic solutions. Whereas sorption of Pb(II) in the presence of Ca(II) and Mg(II) was shown to be slightly affected, Ti_3_C_2_(OH/ONa)_x_F_2−x_ flakes maintain their selectivity toward lead ions with an efficiency of 95.2%. Figure 2c,d shows Pb ion uptake capacity as a function of time and temperature. The lead uptake capacities of these flakes can reach 48 mg/g in 120 s, and they can maintain an average uptake of approximately 55mg/g for up to two hours at room temperature. However, at higher temperatures of 50 °C, uptake of lead ions can reach in excess of 130 mg/g.

Density functional theory (DFT) was used to predict the adsorption behaviors of various MXene nanostructures [71]. N substitution in MXenes was found to be more effective in Pb adsorption compared to C-based kinetics. The most attractive MXene composition for lead removal is Ti_2_C(OH)_2_. The selectivity of MXenes towards metals can be explained with the inner-sphere complexation theory, due to the existence of surface [Ti-O]-H+ groups, in addition to the laminated structure that can help trap metals, which is particularly true for divalent cations, as they have low hydration energies [72,73].

More recently, a new MXene powder modified with a silane coupling agent was proposed as an efficient compound for lead removal [51]. This new compound induces MXene oxidation to produce titanium dioxide, which is then modified with lipophilic unsaturated olefin KH570, thereby acting as a toxic metal reducing agent. This new compound was produced by etching Ti_3_AlC_2_ powder using 40% HF with agitation, followed by two washing periods using water, ethanol, and oxalic acid, with a vacuum drying step in between. The first washing step reduced the pH of the obtained flakes to 6; the second reduced its pH further to 4–4.5. Then, KH570 was added in a water bath at 70 °C with mechanical agitation for two hours to oxidize the surface of MXene flakes and obtain TiO_2_ particles, after which the product was washed with purified water and vacuum dried in an oven at 60 °C to obtain the final product denoted with Ti_3_C_2_T_x_-KH570. Figure 3a provides a schematic representation of the formation of the nanofibers on the surface of an MXene sheet as a result of this process.

Figure 3b–e shows SEM images of the final product with different concentrations of KH570. Interestingly, MXenes reached an adsorption capacity of approximately 150 mg/g at 10% KH570 doping, which was the optimal percentage, with a removal efficiency of up to 98% at pH 5; however, this efficiency drops suddenly as pH levels rise above this value. This compound does not seem to have lead adsorption capacity at pH levels below 3, which allows for particle regeneration in acidic solutions.

Another proposal to increase the number of active sites that can bind to toxic metals is an MXene/alginate composite that increases the adsorption capacity of lead ions to 382.7 mg/g and that of copper ions to 87.6 mg/g [52]. The novel composite was prepared by adding MXene powder to varying amounts of sodium alginate (for 0.16 g of Ti_3_C_2_T_x_, 68, 106, 160, 240, and 640 mg of sodium alginates were tested). The mixture was then stirred in 20 mL deionized water for six hours, and then placed at a low temperature for 12 h before freeze-drying in vacuum for 24 h. The authors used calcium nitrate (0.2 M) as a cross-linking agent on a separate batch. The authors reported that adsorption of Pb and Cu ions increased with increasing ratio of sodium alginate with an optimum MXene/alginate ratio of 70%. Beyond this value, adsorption of Cu ions decreases and adsorption of Pb increases slightly. The optimal operating conditions for this new compound are within the pH range of 5–7, making it suitable for operation in most wastewater treatment conditions. A regeneration study using 0.1 M nitric acid showed that crosslinking MXene alginate with calcium ions increases their recyclability significantly, as the compound can be recycled up to 10 times. After durability cycles, the adsorption rate decreases slightly, with a loss rate of 8.9% for Pb ions and 5.4% for Cu ions. This indicates that crosslinking MXene with alginate can increase the compound’s stability significantly, which produces better and more sustainable performance of the compound over a longer period.

An investigation from the same period presented a novel MXene compound with surface functionalization using three different biosurfactants: chitosan (CS), lignosulfonate (LS), and enzymatic hydrolysis lignin (EHL) [50]. This comparison between the effect of cationic, anionic, and non-ionic surfactants revealed that non-ionic surfactants that serve as obstacles to prevent the restacking of MXene sheets significantly increase the adsorption efficiency of the MXene compound. Thus, EHL-functionalized MXenes displayed an adsorption capacity of 232.9 mg/g, which is far greater than the adsorption capacity of MXene sheets alone or in combination with cationic and anionic surfactants. This compound was prepared by first etching a MAX phase powder in 1 g LiF and 10 mL of 12 mol/L HCl, which was then stirred at 40 °C, and then centrifuged and sonicated in deionized (DI) water. Then, 50 mL of the resulting supernatant was collected and added to 5 mL of 1.0 wt% icy acetic acid, DI water, and 1.0 wt% ammonia that contained 0.1 g of CS, LS, or EHL, followed by stirring for four hours. Afterward, the resulting mixture was freeze-dried for 24 h, and the three new compounds were tested for lead adsorption in comparison with plain MXene powder. Figure 4 illustrates the general morphology of the MXene sheets alone and in combination with CS, LS, and EHL, where the 2D structure of MXene sheets as an agglomeration of few layers is depicted. Maximum adsorption was attained using the Ti_2_CT_x_-EHL compound at pH 5 with a removal rate of 98.9%, after which adsorption values started to decrease. This indicated that non-ionic surfactants increase the adsorption ability of MXene flakes by improving their structure and enhancing the chemisorption of lead ions.

In 2020, Jun et al. presented a practical approach towards the study of MXene adsorption of lead, bringing it a step closer towards its future applicability [49]. This comprehensive exploration of normal MXene flakes outlined the effects of dose concentrations of MXene and Pb(II) separately on its removal. Additionally, the effects of pH variation and background ion and humic acid (HA) levels in the solution on adsorption efficiency were also investigated. Most importantly, the effects of the presence of competing ions, specifically Cu(II), Zn(II), and Cd(II), and the reusability of MXenes in a cyclic adsorption/desorption manner were examined. Interestingly, the preferential absorption of MXene towards the heavy metals followed this order; Pb(II) > Cu(II) > Zn(II) > Cd(II). In addition, MXene was able to be regenerated after absorbing Pb(II), making it possible for re-use if needed.

In comparison with powder-activated carbon, MXene flakes exhibited better performance in the removal of Pb ions, although MXene has a ~50-fold smaller surface area. This indicated that the adsorption process is not physical, but rather chemical and that MXene’s advantage in Pb ion adsorption can be attributed to its negative surface charge. MXene flakes show fast adsorption ability saturated within 30 min with an uptake capacity of ~38 mg/g. This adsorption efficiency of MXenes increases with increasing solution temperature, but not considerably; at pH < 6, it drops significantly. The presence of various salts was shown to have a marked effect on Pb ion adsorption due to screening as well as ion competition between salts of comparable size and Pb. Nevertheless, the presence of HA, which mimics the effect of dissolved organic matter, was shown to slightly increase both adsorption efficiency and uptake capacity of MXenes. This study is important, as it reflects the actual adsorption conditions of wastewater treatment facilities and can be regarded as a template to follow when studying future MXene composites in comparison with basic, unaltered 2D flakes. Notably, in the presence of competition for toxic metals, MXenes show preferential adsorption for lead ions in general; however, their adsorption efficiency tends to drop. Based on the reported results, the uptake capacities of unaltered MXenes for Cu(II), Zn(II), and Cd(II) are about 38, 32, and 3 mg/g, respectively. However, in the presence of a mixture that combines all these toxic metals, MXene flakes show preferential adsorption for lead, ions, but it is significantly lower for other ions. The re-usability of MXene flakes was demonstrated over four cycles using 1 N HCl as a regeneration solution.

In general, surface activation and F reduction seem to be significant enhancement mechanisms to improve Pb ion uptake using Ti_3_AlC_2_ nanoparticles [74]. Although promising results were obtained using this approach, including a 96% removal rate and ~290 mg/g uptake capacity, as these compounds are not part of the MXene family, they will not be discussed further.

### 4.2. Removal and Reduction of Chromium Ions by MXenes

Chromium is among the most toxic metals that are produced regularly in various industries, such as metallurgy, electroplating, leather tanning, and refractory [74]. Chromium ions in wastewater can be present either in a trivalent Cr(III) or a hexavalent Cr(VI) form. Although trivalent chromium is considered a nutrient in aquatic environments at low concentrations with mild toxicity when accumulated in higher concentrations, hexavalent chromium is a known carcinogen. According to the Institute for the Regulation of Water and Solid Waste (IARC), chromium in all its forms can be responsible for asthma, diarrhea, and damage to the liver, kidney, and male reproductive system [75]. Therefore, the maximum allowable Cr(VI) concentration set by the WHO is 0.05 ppm, and the total concentration of chromium in drinking water should not exceed 50 μg/L [67]. This necessitates either reduction of Cr(VI) to trivalent form or complete removal as an essential task in wastewater management.

Based on this study, one of the most promising adsorptive materials is *Lagerstroemia speciose* bark embedded with magnetic nanoparticles with an uptake capacity for Cr(VI) of 739.7 mg/g and a contact time of 0.17–2 h in double distilled water [76]. Electrospun hematite nanofiber, with or without mesoporous silica, functionalized with an amine group (for Cr(III)), calcium ferrite (for Cr(IV)), or manganese ferrite (for total Cr), exhibited an adsorptive capacity of ~340 mg/g at pH 2–6 at room temperature with contact times that range from seconds to two hours [77]. However, most studies listed in the review were conducted in the absence of competitive ions and in distilled water conditions, which does not necessarily reflect their performances under actual wastewater conditions. Most tests were conducted at pH 2–3, which is the optimal range for synthesized particle action; however, industrial wastewater effluents have a pH value of 5 to 8 [43].

Several studies were conducted on the use of MXene flakes for chromium removal or reduction [53]. In this study, the authors immersed Ti_3_AlC_2_ powder in 10%, 25%, and 50% HF solutions. Then, the solution was washed until pH 5–6 was obtained. Sediments were then air-dried over a period of three to four days. The resulting powder was then intercalated in DMSO, followed by washing, ultrasonication, and vacuum drying. The authors showed that MXene sheets etched in 10% HF were thinner, with an average thickness of 3 nm, providing a larger active surface area. These thin layers of exfoliated MXenes were able to maintain their compositions for up to three months before starting to degrade. They exhibited Cr(VI) removal capacity of 250 mg/g due to oxygen-containing bonding sites such as OH and F. However, as the isoelectric point of these nanosheets is about 2.36, their mechanism of action is highly dependent on the pH level of the solution; at pH 2, the surface of the sheets is positively charged due to the abundance of hydroxyl groups with good adsorption ability of negatively charged Cr_2_O_7_^2−^ ions, as shown in Figure 5. However, as pH increases, the electrostatic attraction weakens and slowly degrades until it reaches pH 13, where it stops entirely.

Upon contact with MXene sheets, chromium is reduced with the help of hydrogen ions, and Cr(OH)_3_ starts to precipitate at pH 4.8, reaching full precipitation at pH 5.6, whereas Cr(III) can covalently bond to the surface of the MXene sheets by titanium oxide sites at pH 2 with up to 98% removal capacity. However, these sheets may not be recovered after their use, as they tend to degrade after the adsorption is completed.

Another MXene-derived compound is an urchin-like rutile titania carbon composite [55]. This urchin-like rutile TiO_2_-C (u-RTC) compound was synthesized from Ti_3_AlC_2_ powder by etching in LiF and HCl solution at 60 °C for 48 h. The resulting sediments were washed, followed by in situ decomposition of the obtained MXene in a mixed solution that contained ethylene glycol (EG), FeCl_3_, and isopropyl alcohol (IPA), causing crystalline nanorod formation due to the decomposition of Ti-OH or Ti–F surface bonds into Ti–O groups (TiO_2_) in needle-like formations (Figure 6a). These nanorods were perpendicular to the MXene flakes of average diameter ~180 nm and ~1.6 mm length, as determined from filed-emission scanning electron microscope (FESEM) and TEM images.

The adsorption capacity of Cr on urchin-like rutile TiO_2_-C (u-RTC) nanocomposite outperformed layered anatase TiO_2_-C (l-ATC) (free FeCl_3_, 12 h, 200 °C), u-RTC + l-ATC (FeCl_3_, 24 h, 200 °C), and normal MXene flakes by a large margin (Figure 6b,c). Additionally, the compound shows preferential adsorptive action for Cr(VI) with a maximum sorption ability of ~225 mg/g at pH 3–6 (Figure 6d). The study considered adsorption conditions under the effect of competing ions within wastewater solutions; thus, the removal efficiency was studied in combination as a function of Cl^−^, NO_3_^−^, SO_4_^2−^, F^−^, and PO_4_^3−^ interference, where the compound showed excellent chromium selectivity in general except for phosphate. The product was tested on real wastewater and showed potential. In general, u-RTC sorption of chromium ions is achieved in two stages: the first within the first two minutes, reaching an uptake of 28%, and the second sorption stage spanning 120 min with a total Cr ion removal of 38%. The compound demonstrated a regeneration ability using 5% NaOH for up to five cycles [43].

In 2018, MXene flakes formed using chemical etching HF (40%) solution under magnetic stirring for 24 h were tested for chromium adsorption ability, and they were able to achieve up to 80 mg/g adsorption capacity at room temperature [54]. No data were provided with regard to the effect of pH variations on the adsorption capacity.

The Ti_3_C_2_/TiO_2_ composite achieved fast and efficient removal of Cr(VI) from the potassium dichromate solution, a mechanism that causes instant reduction of Cr(VI) to Cr(III) followed by adsorption of the generated Cr(III) [56]. Thus, after the preparation of MXene flakes using HF as an etchant, the resulting Ti_3_C_2_ was hydrothermally heated to 160 °C to grow Ti_3_C_2_/TiO_2_ particles on the surface. The formation rate of a TiO_2_ on MXene sheets can be adjusted by varying hydrothermal treatment time. This study revealed that 24 h of hydrothermal treatment provides optimal Cr(VI) removal capacity, with a reduction efficiency of 99.35% within 12 min under acidic conditions. Figure 7a presents a schematic of the Cr(VI) reduction and removal mechanism using these MXene sheets. Figure 7b presents an SEM element mapping of the composites before and after chromium adsorption at two different pH levels, 2 and 5. However, no clear adsorption capacity in terms of mg/g was presented.

A novel approach to the enhancement of MXene adsorption of chromium ions was reported, where nanoscale zero-valent iron (nZVI) particles were intercalated into the inner layers of alk-MXene sheets [57]. The main aim of this new compound was to increase the interlayer spacing of MXene flakes using nZVI molecules. Alkalization treatment was performed to increase the number of active sites available on the interlayer surface. The authors showed that the uptake capacity of nZVI-alk-Ti_3_C_2_ could reach 194.87 mg/g, which is significantly higher than that of pristine Ti_3_C_2_, which is 30.6 mg/g. Maximum adsorption occurs in highly acidic solutions, pH 2, after which the efficiency drops significantly. This is an obvious drawback of this compound in toxic metal remediation applications, despite the product showing clear selectivity of chromium ions in the presence of competing ions.

The most impressive strengths of MXene compounds are their flexibility and ability to bind with various particles, significantly affecting their properties to favor certain applications. Such modifications were attempted using polymerization of MXene sheets, which increased the adsorption efficiency and reduction of hexavalent chromium [58]. Polymerization of MXene sheets is accomplished in situ by intercalation of poly(m-phenylenediamine) (PmPD) into regular MXene sheets. The process is straightforward: 1 g of mPD is dissolved in 30 mL DI water, which is then added to 100 mL of MXene dispersion (Ti_3_C_2_T_x_/PmPD-X mass ratios of 2:1, 5:1, and 10:1 were tested). The solution is then sonicated for 30 min, after which Na_2_S_2_O_8_ solution is slowly added. The mixture is left to react for four hours at −4 °C. Then, the solution is centrifuged, rinsed with DI water, and vacuum dried at −55 °C. The morphology of the obtained modified 2D composite is shown in Figure 7c–f, where SEM images of normal MXene sheets are shown together with Ti_3_C_2_T_x_/PmPD-2/1, Ti_3_C_2_T_x_ /PmPD-5/1, and Ti_3_C_2_T_x_/PmPD-10/1. The effect of polymerization is evident with the increase of PmPD ratio, accompanied by a marked augmentation in the 2D structure thickness. The maximum removal ability of Cr(VI) was achieved using Ti_3_C_2_T_x_/PmPD-5/1 with an adsorption capacity of 540.47 mg/g in comparison with 384.73 mg/g of PmPD and 137.45 mg/g of MXene. This indicated that the augmentation of the performance of this new compound could be attributed to the synergistic effects of both MXene and PmPD. Figure 7g shows the adsorption mechanism of the compound, where anionic hexavalent chromium is adsorbed; then, about 51.6% is converted to Cr(III) by the benzenoid amine, which in turn is oxidized into quinone imine. This is followed by Cr(III) chelation onto the protonated quinoid imine of the composite. Despite the impressive performance of this compound, its optimal adsorption capacity peaks at extreme acidic conditions (pH 2), after which its performance decreases markedly, reaching a removal rate of ~30% at pH 6, and losing its adsorption capacity at pH 9. Thus, the composite can be regenerated in an alkaline solution. As such, a regeneration study was performed using a NaOH solution, and the efficiency of the compound remained at ~90% after five cycles.

### 4.3. Copper Ion Removal from Wastewater

Copper ion, Cu(II), is another toxic metal being increasingly disposed of in industrial wastewater as a result of various manufacturing processes, including fertilizer production, mining, battery production, electronics production, pharmaceutical production, and paper manufacturing [78]. Although low concentrations of copper are used to control bacterial growth in lake water and prevent biofouling, for human physiology, limited concentrations of copper compounds are regarded as a necessary part of the human daily diet. Acute copper exposure can lead to kidney and liver failure, gastrointestinal bleeding, intravascular hemolysis, and hematuria; at lower doses, copper poisoning symptoms are similar to that of food poisoning, including headache, nausea, vomiting, and diarrhea. Therefore, according to the EPA, the maximum copper concentration in drinking water should not exceed 1.3 ppm [79]. Copper adsorptive materials are vast and span biological and industrial products from the macro to the nanoscale.

Recently, various adsorptive agents, such as modified natural materials, industrial by-products, modified agricultural and environmental waste, biosorbents, and mesoporous carbon, were outlined [78]. According to this review, natural material adsorption of copper can reach a maximum of 83.3 mg/g using HCl-treated clay. Microbial biomass, and more specifically, lignocellulosic materials such as modified orange peel, can reach an uptake capacity of 289 mg/g. Graphene-oxide-based nanomaterial showed an average adsorption capacity ranging from 18 to 425 mg/g with a maximum experimental adsorption capacity of 588 mg/g at pH 5.

Several studies have been conducted for the removal of copper using MXenes. Exfoliated MXene nanosheets prepared using ultrasonication in degassed and deionized water under a nitrogen gas flow at 60% amplitude with a pulse rate of 3/1 s significantly enhanced the removal efficiency of copper [59]. The adsorption action of this compound occurs at oxygenated moieties on the surface, which facilitates the reduction of Cu(II) ions, forming Cu_2_O and CuO species. The authors demonstrated that the delamination of MXene sheets increases the adsorptive ability up to 78.45 mg/g with a fast absorbance rate that is capable of removing 80% of the total content in less than one minute and continues to rise to about 97%. Adsorption of Cu ions is accompanied by surface oxidation of MXene flakes before and after the adsorption of copper. The morphological changes on the MXene surface due to TiO_2_ particle formation during adsorption of Cu are clearly visible. The effect of pH on Cu adsorption was also investigated; as expected, Cu adsorption is pH-dependent with low adsorption rates at low pH due to reduced surface charge. The point of zero charges of these MXene flakes is at pH 2.7. Beyond this value, the surface becomes negatively charged and starts performing as a metals adsorbent. Optimum adsorption capacity is reached at pH 5, after which adsorptive action starts to decline due to precipitation of Cu^2+^ ions into Cu(OH)_2_ in the aqueous phase. Since the surface charge of these flakes becomes positive at low pH, regeneration is possible in acidic solutions. The authors used nitric acid and calcium nitrate for five hours. These flakes did not show good regeneration ability, as their adsorption performance declined exponentially from 80% in the first cycle to 47% and 30% in the second and third cycles, respectively, due to surface oxidation and formation of TiO_2_ particles.

More recently, hydrochar-wrapped MAX-phase-derived nanofiber composites were synthesized by mixing Ti_3_AlC_2_ particles in a NaOH aqueous solution under stirring for three hours in ambient conditions followed by addition of either glucose, cellulose, or pinewood sawdust with different concentrations, and then autoclaved [80]. This treatment transformed Ti_3_AlC_2_ into nanosheets, nanofibers, or bulk particles based on the hydrothermal conditions of the experiment: the NaOH concentration and temperature. Therefore, to obtain nanofibers, 10 mol/L NaOH solution was used at 180 °C for 48 h under static conditions. This was followed by hydrothermal treatment. Once cooled, solid particles were filtered and washed with deionized water until pH 7–9 was reached, and then samples were washed with ethanol and dried at 80 °C for 12 h. The resulting Ti_3_AlC_2_-derived nanofibers (TNFs) combined with glucose, cellulose, or pinewood sawdust were then tested for their adsorption capacities for copper and cadmium ions. This study is interesting because MAX particles were combined with other adsorptive agents with proven toxic metal adsorption capacities. The findings showed that these compounds have a better toxic metal adsorption capacity in comparison with untreated MAX flakes and regular TNFs, as shown in Figure 8. These combinations can be tested using MXene sheets, as the delaminated surfaces of the flakes might significantly add to the adsorptive potentials of the compounds. In this study, the optimal adsorptive action of the compound occurred at pH 6, where the zeta potential reached −10.9 mV. Regeneration of the particles was successful in an acidic solution of 0.2 mol/L HCl with good regeneration ability of glucose/Ti_3_AlC_2_ (Glu@TNFs); however, the durability was not investigated.

The adsorption abilities of these particles are shown in Figure 8b (for both copper and cadmium). In general, cellulose-infused TNFs have better adsorption compared with other compounds, with a capacity of ~64 mg/g for Cd ions and ~42 mg/g for Cu. Although these results are still modest in comparison with other experimental nano-adsorbents, these composites are promising, especially regarding the possibility of increasing MXene stability and reusability. However, further studies are necessary to explore the full potential of such an approach using MXene.

MXenes surfaces functionalized with amino acids showed effective performance for copper removal [60]. This MXene-based polymeric composite can be prepared under mild settings, such as room temperature, alkaline conditions, and without the need for any catalysts, in a single-step procedure. The surface modification technique is based on the strong adhesive properties of catechol that tends to self-polymerize under alkaline conditions, forming a functional coating material. Although the manufacturing concept is impressive, the experimental results obtained using the compound were modest, with an adsorption capacity of 46.6 mg/g in alkaline settings. The compound was not tested with other toxic metals to discern its ability to attract other toxic metals. The manufacturing of this particular compound was achieved by adding MXene powder (0.7 g) to a Tris buffer solution (20 mL, 10 mmol/L, pH 8.5) combined with ultrasonic agitation for 10 min. Levodopa (DOPA) (0.8 g), an amino acid that contains catechol and carboxyl groups, was dissolved in diluted hydrochloric acid (5 mL), and the pH was adjusted to 8–9 using Tris buffer solution. Then, DOPA solution was then added to the MXene suspension under magnetic stirring with a pH around 8.5. The mixture was stirred for eight hours, and the final product was centrifuged, washed, and dried. This compound has an optimal adsorption capacity at pH 11 with a removal efficiency of 93.2%, beyond which its performance starts to decline. No regenerative studies were conducted to discern compound stability and recyclability. This composite helps emphasize the importance of the surface charge effect of a MXene on its toxic metal adsorption capacity; therefore, increasing surface area alone may not help improve the adsorption capacity, particularly since wastewater confluents are mostly acidic or neutral in nature rather than alkaline.

### 4.4. Mercury Removal Using MXene Composites

Mercury pollution is a global problem caused by burning coal or waste in addition to other industrial and agricultural processes, such as cement production, mining, paper production, and battery production [81]. Mercury ions are toxic, volatile, and tend to accumulate, causing severe health issues. Whereas the noxious effect of mercury on the nervous system and neurons is well known, in biological entities, mercury can react with amino acids containing sulfur as well as albumin and hemoglobin, which merits its ranking as one of the top 10 most toxic metals. Therefore, the maximum Hg level in drinking water is 0.001 mg/L, according to WHO [82]. Successful adsorption of mercuric ions is achieved using various materials, such as structured porous organic polymers, fly ash, and various others [83]. In general, the average adsorption capacity of experimental compounds is around 100 mg/g, with lower ones being around 11 mg/g [84], and the maximum adsorption capacity reaches about 769 mg/g [85]. Although composite compounds that combine conducting polymers, such as polyaniline, with reduced graphene oxide, have shown tremendous promise with adsorption capacity values that reach up to 1000 mg/g [86], three studies have been conducted so far on MXene mercury adsorption [61,63]. This included testing the modified MXene flake adsorption ability of mercuric ions, producing excellent results on a wide range of pH conditions with a maximum Hg(II) uptake of 1128.41 mg/g [61]. This was achieved by magnetizing MXene flakes as a means to increase their stability and provide a fast recovery mechanism of the compound once adsorption was completed (Figure 9a). Therefore, functional groups of Fe_2_O_3_ and Fe_3_O_4_ nanoparticles were deposited on the surfaces of MXene particles (Figure 9b). To achieve this, once Ti_3_C_2_T_x_ nanosheets were synthesized, flakes were suspended in a solution of deionized water containing 0.25 mg/mL iron acetate (Fe(C_2_H_3_O_2_)_2_). The solution was then continuously stirred under Ar gas for 30 min. Afterward, the suspension was autoclaved after the addition of 100 μL of ammonia, which acted as a catalyst, and heated at 180 °C for four hours. The hybrid nanocomposite powder was centrifuged, washed with water/ethanol, and freeze-dried. Then, the dried powder was thermally annealed at 400 °C for four hours.

The obtained magnetic titanium carbide (Ti_3_C_2_T_x_) MXene (MGMX) nanocomposites were tested for Hg(II) adsorption using HgCl_2_ in 5% HNO_3_ distilled water. Solution pH was adjusted to test its efficacy within a pH range from 2 to 9 at room temperature. At a high acidic pH value of 2, Hg(II) adsorption was very low, with 18.7% removal efficiency, whereas maximum adsorption was achieved at pH 6 with 47.46 mg/g and removal efficiency of 98.89% (Figure 9c,d). The compound was tested for ion selectivity in the presence of Mg(II), Na(I), Ca(II), and K(I), and a Hg(II) removal capacity >98.95% was achieved. This selectivity, according to the authors, is dependent on the abundance of various functional groups on the surface of MXene sheets, which allows for mercuric ions to combine with oxygenated and hydroxyl moieties. The compound showed excellent recyclability, as it was able to maintain its adsorption capacity for up to four cycles; its adsorption capacity was reduced to 81% in the fifth cycle.

In the second study, MXene core-shell spheres were used for mercuric ion removal [62]. This compound was synthesized using ammonium fluoride as an etching agent by immersing MAX flakes into 100 mL of 1 M NH_4_F solution at room temperature for 24 h. Then, the mixture was washed until pH 6–7 was achieved; afterward, the filtrate was dried in a vacuum. The obtained MXene flakes were ultrasonicated for 10 min in deionized water under Ar gas with 20 mg/mL sodium alginate (SA) powder, which is a naturally occurring biopolymer that has hydrogel characteristics with multivalent cations. This was followed by thorough dispersion by magnetic stirring. Then, aqueous MXene-alginate mixture was injected as small droplets using a syringe needle into a CaCl_2_ solution. These spheres were stirred magnetically again for one hour followed by rinsing until pH 6–7 was attained and freeze-dried resulting in spherical compounds ~3 mm in diameter in wet form and ~2 mm in diameter in dry form (Figure 10a–d). SEM imaging of the dried sphere sections revealed a core-shell structure that is an amalgamation of MXene-alginate (Figure 10e–g).

The kinetics and isotherms of Hg(II) adsorption are shown in Figure 10h,i. These spheres exhibited outstanding adsorption ability of Hg(II), which reached 932.84 mg/g, with up to 100% efficiency in comparison with 34.63% and 11.53% for sodium alginate alone and graphene oxide sodium alginate spheres, respectively. The investigation involved a study in optimizing the MXene to sodium alginate ratio, and increasing MXene concentration ratio of 4 to 20 showed optimal results with 90% ion diffusion capacity in 60 min and complete removal in 120 min. Regeneration ability was examined, and an 8 M HCl concentration was used to achieve 99.69% desorption of Hg ions. The extreme acidic conditions needed for desorption are due to the wide range of adsorption capacities of these spheres that span a range of pH 2–11. Multiple element performance showed that the compound was able to achieve 90% removal capacity of five other toxic metals, Cr, Co, Ni, Cu, and Zn, in addition to the removal of 80% of Pb, Hg, and Cd ions. The adsorption mechanisms of these spheres can be attributed to several mechanisms, including complex formation between [Ti–O]–H^+^ and Hg^2+^, ion exchange between Ca^2+^ and Hg^2+^, and electrostatic interaction of the porous surface that is abundant with hydroxyl, alkane, and carboxyl groups [81]. The unique fabrication method of this compound with the results obtained presents a promising solution for future MXene compounds, particularly given the stability increase of these spheres in a variety of pH conditions.

In their most recent investigation, a new MXene compound was reported for mercury ion removal based on molybdenum-disulfide functionalized MXenes (MoS_2_/MX) [63]. The synergistic action of sulfur and oxygenated MXene surface termination helped achieve a maximum Hg ion adsorption capacity of 1435.2 mg/g. The synthesis of this new compound is shown in Figure 11a, which can be summarized as follows: Both multilayered and delaminated MXene sheets were combined with 0.88 mM thioacetamide (CH_3_CSNH_2_) and 9.2 nM sodium molybdate (Na_2_MoO_4_.2H_2_O) under magnetic stirring. After a reaction period of one hour at 30 °C, the mixture was transferred to a Teflon-lined autoclave and maintained for 12 h at 180 °C, followed by the delamination of MXene-based compounds (MoS_2_/MX-I) and (MoS_2_/MX-II). The SEM images of both MoS_2_/MX-I and MoS_2_/MX-II compounds in comparison with regular MXene flakes are shown in Figure 11b–e, and elemental mapping in MoS_2_/MX-II along with the elemental content in Figure 11f,g. Mercuric ion removal of MoS_2_/MX-II was higher in comparison with stacked and typical MXene flakes with a removal efficiency of 98.5% (compared with 74.8% of normal MXene), and a faster adsorption action that can reach an equilibrium in two minutes. The MoS_2_/MX-II compound presented a high adsorption efficiency over a wide range of pH values (2–11), which indicated the remarkable stability of the new compound as well as its ability to function under various environmental conditions.

The regeneration of this compound is particularly challenging. Researchers used 5 M HCl for desorption with a contact time of 30 h to achieve 55% desorption. MoS_2_/MX-II showed remarkable selectivity for (Hg^2+^, Cd^2+^, and Pb^2+^) adsorption relative to (Na^+^, Mg^2+^, Ca^2+^, and K^+^), but with preferential adsorption of Hg^2+^.

### 4.5. Barium Extraction Using MXenes

Barium ions are regularly generated in industrial waste due to various chemical, photochemical, and metallurgic processes that are associated mainly with toxic industries such as the automotive and petrochemical industries [87]. In addition to their high solubility in water, barium ions have the potential to spread to larger aquatic areas rather than stay concentrated at their discharge site, making their disposal particularly dangerous in natural aquatic bodies without proper treatment [88]. Barium doses as low as 0.2–0.5 g have toxic effects in adult humans, with quantities as high as 3–5 g considered lethal. Barium poisoning causes gastroenteritis and muscular paralysis that is attributed to hypokalemia. Therefore, the maximum allowable concentration of barium is set by the WHO to 1 mg/L in the U.S. and Canada in drinking water and 0.1 mg/L in surface water [89]. Barium removal has been investigated by gas production companies as well as others to remove barium ions generated by the fracking industry. Such methods generally rely on sulfation. Strontium sulfate co-precipitation with barium that can achieve 97–100% removal of barium ions [90]. Natural organic adsorbents, such as *Aloe vera* waste, were shown to have an adsorption capacity of up to 107.5 mg/g for barium [91]. Alternative methods of barium removal that rely on photocatalysis were reported [87]. This approach is particularly of interest in this review as it relies on TiO_2_ for barium ion reduction. Although this approach was able to achieve a 50% reduction in barium ion concentration after 120 min, using anatase, the photocatalytic effect can be optimized to remove barium ions (99%) [92]. Therefore, the integration of photocatalytic principles of anatase into the design of MXene compounds may yield effective results. To date, only one study was published in 2017 with regards to barium adsorption using MXene compounds [64]. In this study, traditional MXene flakes were synthesized with HF solution, which was then washed and dispersed in ethanol with ultra-sonication under argon, then freeze-dried overnight. The adsorption ability of this pure MXene compound was shown to be pH-dependent (Figure 12a), with an optimum pH of 7 and 100% removal efficiency. The adsorption value began to decrease at higher pH values, and at values beyond 9, barium metal precipitation occurred, resulting in metal hydroxide salt. Since the compound becomes neutrally charged at pH 2.4, regeneration is possible in highly acidic solutions. The adsorption mechanism of this compound is linked with fluoride and hydroxide functional group of the MXene surface in addition to the adsorption effect of the intercalated and un-stacked MXene layer (Figure 12b). Multi-metal adsorption tests were performed in a solution that included Pb, Cr, Ca, Sr, and As ions. The compound showed high selectivity for barium compared to other coexisting metals with a removal capacity reaching 98%. In general, although this compound showed a limited barium removal capacity of 9.3 mg/g, it provides an ideal baseline for future studies of modified MXene flakes to benefit from the additional properties of these compounds.

### 4.6. Phosphate Removal with MXenes

Phosphate is an essential nutrient for plants, but too much phosphate causes eutrophication, a process that starts with the increased and uncontrollable growth of algae in aqueous bodies of water, decreasing other aquatic life forms, followed by the accumulation of organic substance as a product of dead algae and excessive consumption of oxygen to degrade the resulting organic material. This leads to the creation of an anaerobic environment that destroys most aquatic lifeforms due to loss of oxygen, in addition to bad odor and appearance. These changes degrade the quality of water sharply, making it unsuitable for human or animal consumption. Concentrations as low as 0.01 mg/L phosphorus are enough to start eutrophication under the right circumstances [93,94]. Today, metal oxides and metal hydroxides are among the most promising phosphate adsorbents with adsorption capacities of 3.5–345 mg/g and 9.8–514.2 mg/g, respectively [95]. MXene flakes were found to adsorb up to 2400 mg/g of phosphate. This impressive adsorption capacity is due to the unique ferric oxide morphology of these specially synthesized MXene flakes [65].

The synthesis of these individual nanoparticles followed the regular etching of MAX phase powder using HF, a solution of FeSo_4_ and FeCl_3_ was added to MXene flakes and maintained at 80 °C for four hours, after which the mixture was added to a boiling solution of NaOH for 10 min under constant stirring. Then, the solution was left at 100 °C for five hours to allow for crystal formation of MXene–iron oxides MXI. By using this method, iron oxide particles can be as small as 18 nm on the MXene surface. Thus, the main MXI structure is mainly composed of MXene with large amounts of magnetite–Fe_3_O_4_ attached to its surface, and a small amount of maghemite–Fe_2_O_3_ mostly intercalated into its matrix [87]. A phosphate sorption test was conducted at various pH levels and found to be optimal within pH 2.5 to 6, as higher levels of pH cause a dramatic decrease in phosphate adsorption. This is mainly due to most phosphate species existing in negatively charged particles such as HPO_4_^2−^ and H2PO_4_^−^, whereas the zero potential charge on the MXI surface is found to be around pH 6.82. This restricts the sorption ability of these flakes in acidic conditions. Selectivity in the presence of competing ions was tested using one competing ion at a time. These included SO_4_^−2^, NO^3−^, and Cl^−^ ions, where MXI showed excellent selectivity toward PO_4_^3−^, with removal rates of ~72%, ~80%, and ~78%, respectively. Phosphate removal from contaminated water samples taken from various locations was also tested, demonstrating the potential of these compounds with regards to phosphate adsorption in actual wastewater conditions. Although the product can be magnetically separated from the fluid and regenerated in alkaline solutions, no cyclic test was performed to examine the number of usable cycles.

### 4.7. Limitations of MXenes Regarding Toxic Metal Removal

Despite the great advances in the fabrication of MXenes, there are various limitations for the utilization of MXenes in the removal of toxic metal from wastewater, including the inferior durability, lesser-biocompatibility, high tendency for aggregation, and improper reusability [9]. Meanwhile, the rational synthesis of MXene with well-defined and controllable surface chemistry remains a grand challenge. Although there are more than 30 types of MXene compounds, most of them were predicted theoretically, so more attention is needed to synthesis functionalize different types of MXene experimentally suitable for water treatment. Additionally, the cytotoxicity and biocompatibility of MXenes should be investigated deeply prior to their implementation in wastewater treatment. Noticeably, most previous studies focused only on Ti_3_C_2_Tx (T = F, O, and OH) for water purification without enough emphasis on other MXenes. However, Ti_3_C_2_Tx MXene possesses small interlayer spacing and tends to restack in aqueous solutions, which limits its large-scale application in removing toxic metals with large hydrated ionic radii. The stability, reusability, and surface accessibility of Ti_3_C_2_Tx and other MXenes for removal of toxic metals from wastewater could be improved significantly via recombination with low-cost, abundant, and stable polymers, carbon-based materials with high surface areas and unique physicochemical properties [96,97,98,99,100]. From our point of view, the utilization of MXene nanoarchitectures in removing toxic metals and hazardous pollutants from real wastewater should be studied deeply. Additionally, the comparison between MXenes and other adsorbents such as carbon nanotubes, metal-organic frameworks (MOF), and graphene should be highlighted more.

## 5. Conclusions and Future Prospects

In summary, MXene nanoarchitectures are highly promising adsorbents for the removal of toxic metals from wastewater. However, in the future, more attention should be dedicated to the rational design of novel MXene nanoarchitectures experimentally with multiple transition metals as well as controllable surface chemistries and properties to trap toxic metals. Additionally, future work should focus on tailoring the crystallinity, interlayer spacing, and other surfaces/bulk properties for improving the wastewater treatment. More studies are needed on the utilization of different MXene nanoarchitectures; for example, the so-called lightweight MXene (Ti_2_CTx) instead of the heavier Ti_3_C_2_T MXene that is widely used in toxic metal removal from wastewater. The stability and reusability of MXenes are essential issues in the future large-scale wastewater treatment applications. The mechanism of toxic metals and other pollutants should be studied experimentally under real reaction conditions compared to various commercial adsorbents, such as graphene, onion-like carbons, carbon nanatotubes, diamond, and MOF-derived carbons.

In conclusion, this review summarized the recent advances in the preparation and characterization of MXenes for toxic metal removals from wastewater. We included the removal of Pb, Cu, Zn, and Cr, along with their environmental impacts and adsorption mechanisms on MXene nanostructures. The limitations of MXenes’ nanostructures in wastewater treatment were also discussed.

## Figures and Tables

**Figure 1 nanomaterials-10-00885-f001:**
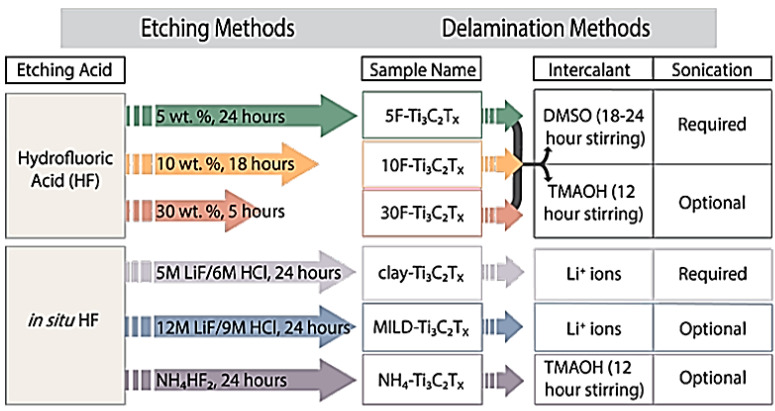
Preparation of Ti_3_C_2_T_x_ using different synthesis routes (direct HF and in situ HF). Reproduced with permission from [36]. Copyright 2017 ACS.

**Figure 2 nanomaterials-10-00885-f002:**
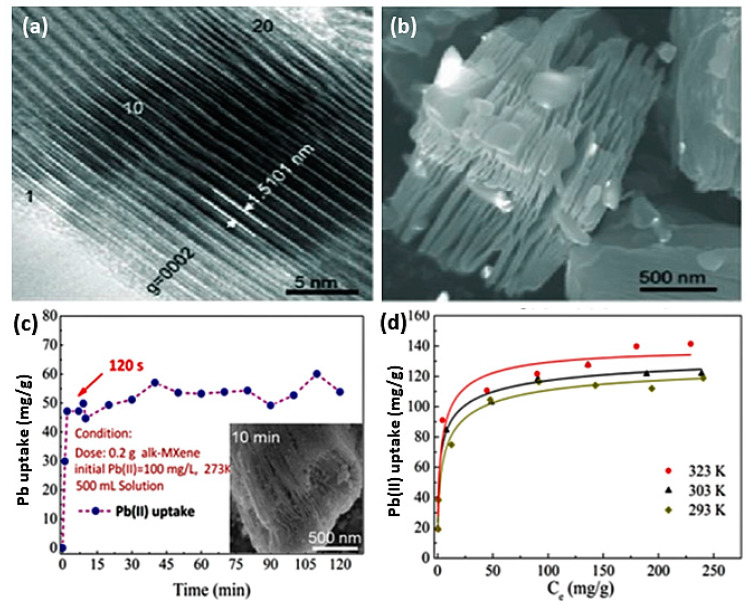
(**a**) SEM graph of alk-MXene after 10 h of exfoliation (HF solution). (**b**) HRTEM micrograph of alk-MXene. (**c**) Kinetics tests and SEM morphology of lead sorption. (**d**) Temperature-dependent lead sorption isotherms onto alk-MXene. Adapted and reproduced with permission from [48]. Copyright 2014 ACS.

**Figure 3 nanomaterials-10-00885-f003:**
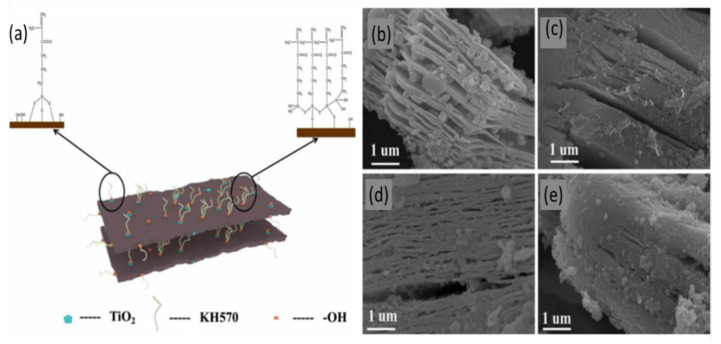
(**a**) Synthesis of Ti_3_C_2_T_x_-KH570 composite. SEM images of Ti_3_C_2_T_x_–KH570 composite with the different KH570 dosages of (**b**) 5%, (**c**) 10%, (**d**) 15%, and (**e**) 20%. Adapted and reproduced with permission from [51]. Copyright 2019 Springer.

**Figure 4 nanomaterials-10-00885-f004:**
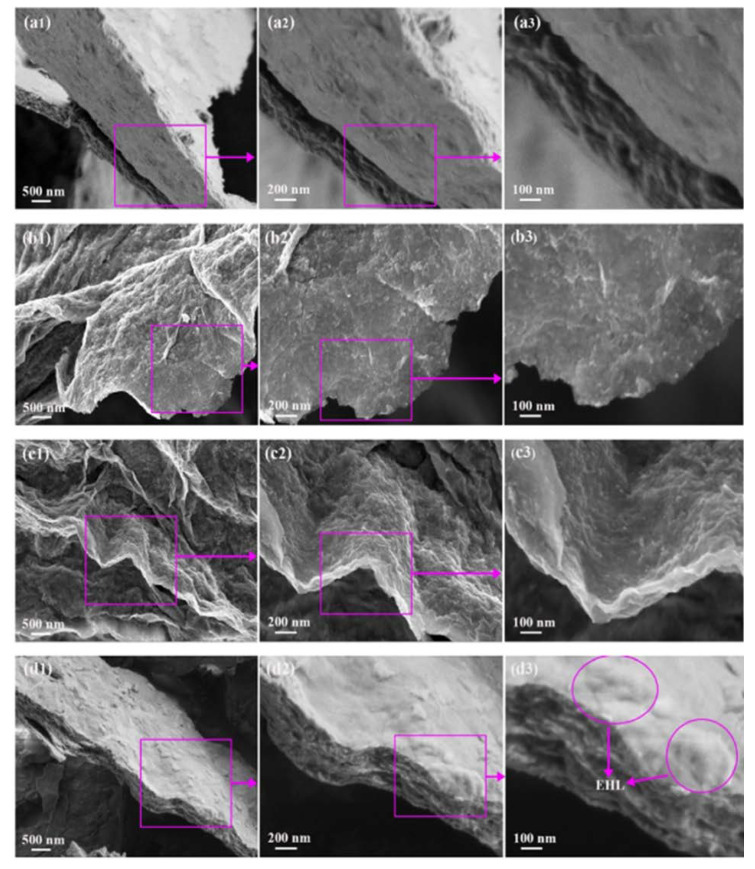
Surface morphologies of (**a1**,**a2**,**a3**) MXene sheets, (**b1**,**b2**,**b3**) MXene sheets–chitosan, (**c1**,**c2**,**c3**) MXene sheets–lignosulfonate, and (**d1**,**d2**,**d3**) MXene sheets–enzymatic hydrolysis lignin. Reproduced with permission from [50]. Copyright 2020 Elsevier.

**Figure 5 nanomaterials-10-00885-f005:**
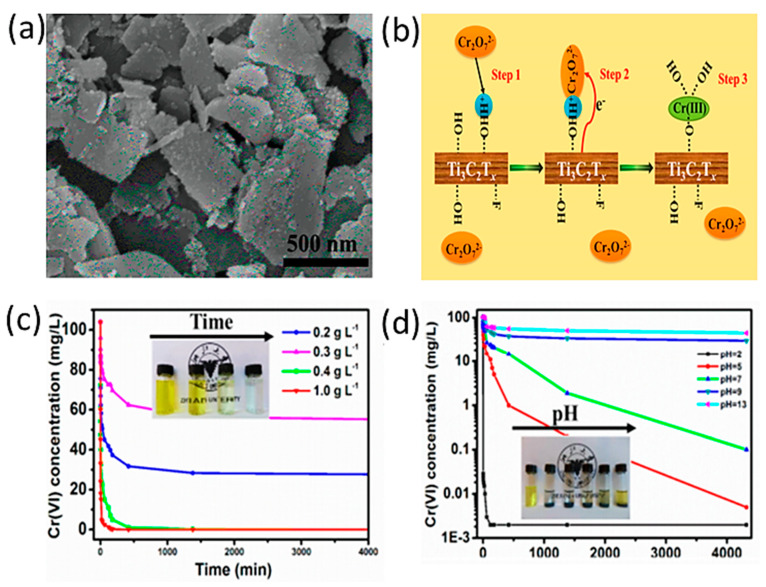
(**A**) SEM image of exfoliated Ti_3_C_2_T_x_ nanosheets (HF 10%). (**B**) Mechanism of Cr(VI) removal by Ti_3_C_2_T_x_. (**C**) Effect of varying Ti_3_C_2_T_x_-10% nanosheets on Cr (VI) concentration (pH 5.0). (**D**) Effect of varying the pH on the removal of Cr (VI) (Ti_3_C_2_T_x_-10%, nanosheets concentration: 0.4 g L^−1^). Adapted and reproduced with permission from [53]. Copyright 2015 ACS.

**Figure 6 nanomaterials-10-00885-f006:**
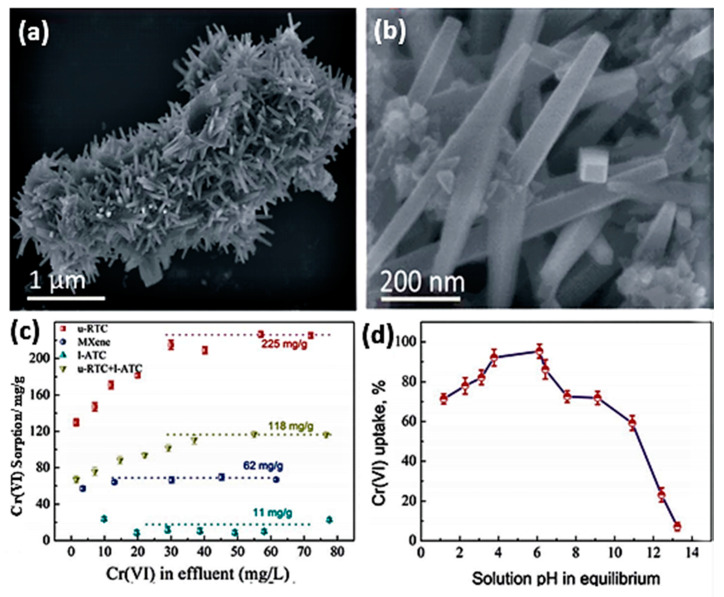
(**a**) SEM image of the u-RTC composite prepared in (MXene-IPA0.5 EG)-1 FeCl_3_ (12 h). (**b**) Local high magnification of u-RTC. (**c**) Cr(VI) sorption performances on the prepared l-ATC in (MXene-IPA)-0.5EG; u-RTC and lATC in (MXene-IPA-0.5EG)-1 FeCl_3_; and u-RTC in (MXene-IPA-0.5EG)-1 FeCl_3_ and the primitive MXene. (**d**) Effect of pH on Cr(VI) uptake by u-RTC. Adapted and reproduced with permission from [55]. Copyright 2016 RSC.

**Figure 7 nanomaterials-10-00885-f007:**
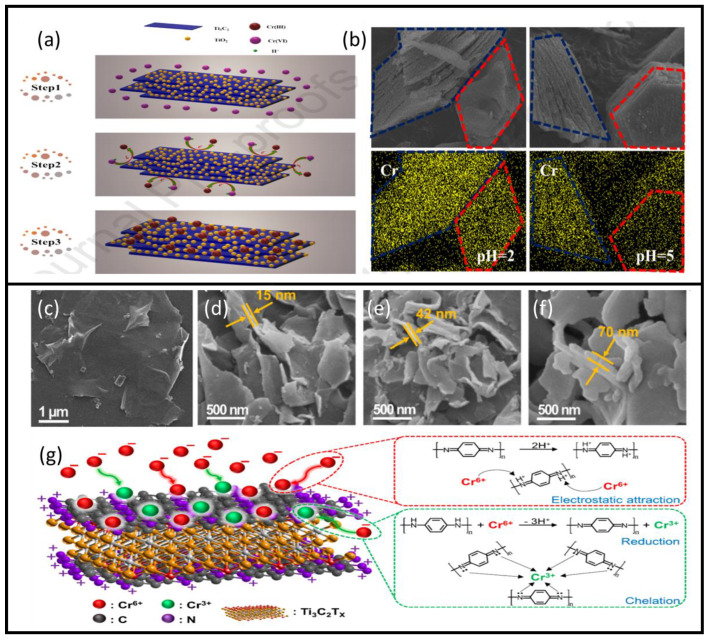
(**a**) Cr(VI) reduction and removal mechanism using Ti_3_C_2_/TiO_2_ composite. (**b**) SEM element mapping of the Ti_3_C_2_/TiO_2_ composites before and after chromium adsorption at two different pH levels: 2 and 5. Adapted and reproduced with permission from [56] Copyright 2020 Elsevier. SEM images of (**c**) Ti_3_C_2_T_x,_ (**d**) Ti_3_C_2_T_x_/PmPD-2/1, (**e**) Ti_3_C_2_T_x_/PmPD-5/1, and (**f**) Ti_3_C_2_T_x_/PmPD-10/1, and (**g**) Ti_3_C_2_T_x_/PmPD adsorption mechanism of Cr (VI). Adapted and reproduced with permission from [58]. Copyright 2019 MDPI.

**Figure 8 nanomaterials-10-00885-f008:**
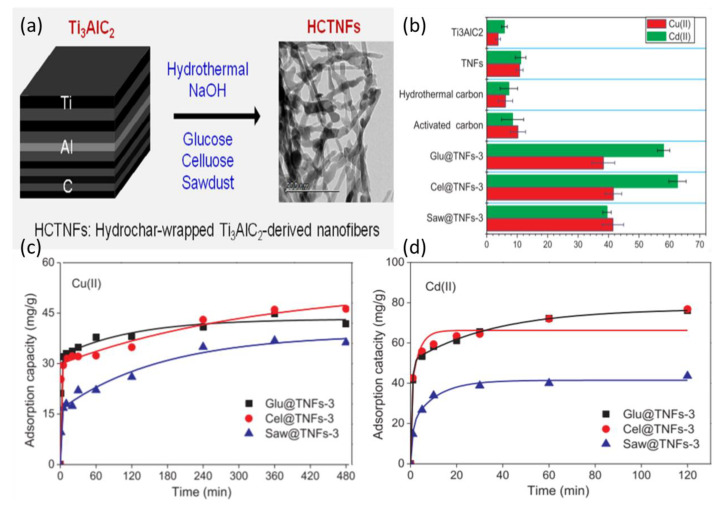
(**a**) Preparation of hydrochar-wrapped, Ti_3_AlC_2_-derived nanofibers (HCTNFs). (**b**) Adsorption capacities of Ti_3_AlC_2_, TNFs, hydrothermal carbon, activated carbon, and typical HCTNFs for Cu(II) and Cd(II). Effect of contact time on the adsorption capacities of Glu@TNFs-3, Cel@TNFs-3, and Saw@TNFs-3 for (**c**) Cd(II) and (**d**) Cu(II). Conditions: pH of Cd = 6, pH of Cu = 5, with initial concentration 100 mg/L for each metal, m/V = 1 g/L, and T = 25 °C Adapted and reproduced with a permission from [80]. Copyright 2019 Elsevier.

**Figure 9 nanomaterials-10-00885-f009:**
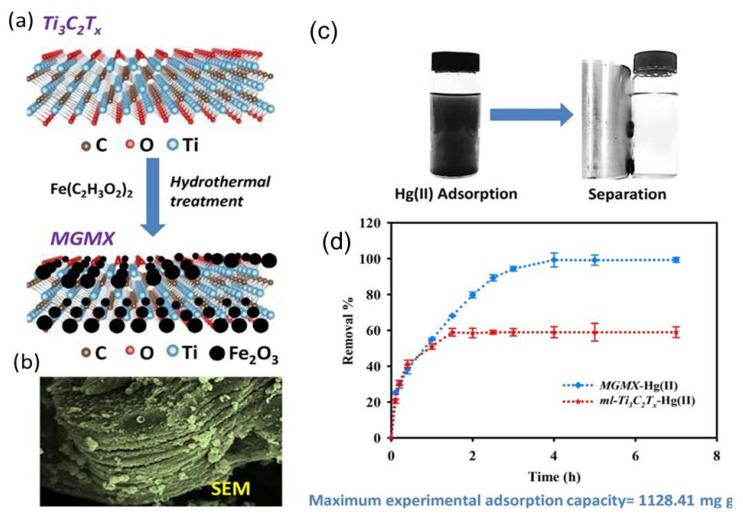
(**a**) Fast recovery mechanism after adsorption. (**b**) SEM image of magnetic titanium carbide (*Ti_3_C_2_T_x_)* MXene (MGMX) nanocomposite. (**c**) Photo for the magnetic separation adsorption efficiency of Hg(II) by ml–Ti_3_C_2_T_x_ and that of the MGMX nanocomposite, and their removal efficiencies (**d**). Experimental conditions: 10 mg/L of Hg(II), 0.025 g/L of adsorbent at pH 6, and 298 K. Reproduced with permission from [61]. Copyright 2018 Elsevier.

**Figure 10 nanomaterials-10-00885-f010:**
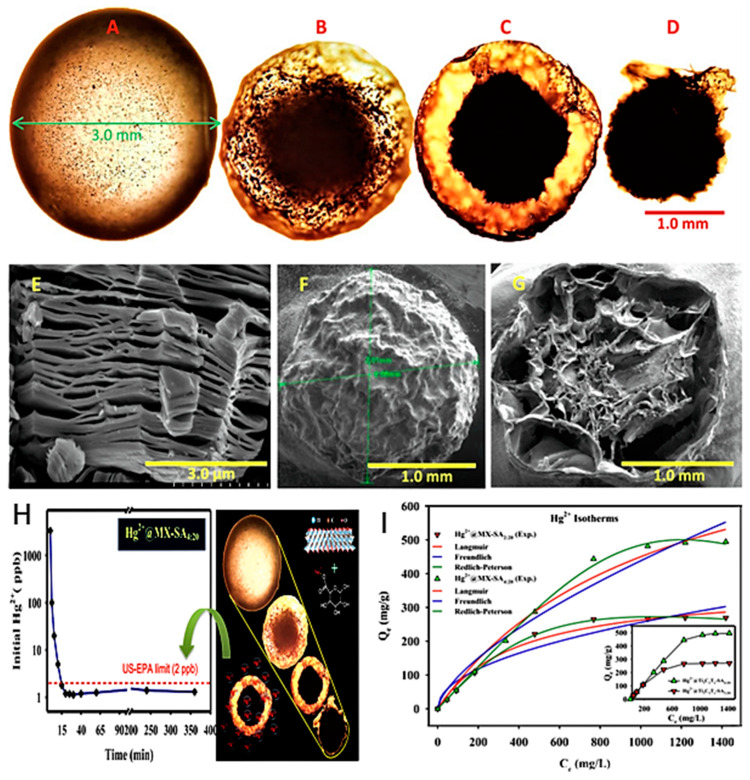
Optical microscopic images of an MX-SA4:20 (**a**) hydrogel sphere (3 mm), (**b**) following vacuum freeze-drying (~2 mm), (**c**) cross-sectional view, and (**d**) core of MX-SA4:20 spheres. SEM images of (**e**) Ti_3_C_2_T_x_ MXene, (**f**) dried MX-SA4:20 sphere, and (**g**) internal structure of sphere cross-section. (**h**) Kinetics of removal of Hg^2+^ from tap water by MX-SA4:20. (**i**) Adsorption isotherm of Hg^2+^ onto MX-SA2:20 and MX-SA4:20. Adapted and reproduced with permission from [62]. Copyright 2019 Elsevier.

**Figure 11 nanomaterials-10-00885-f011:**
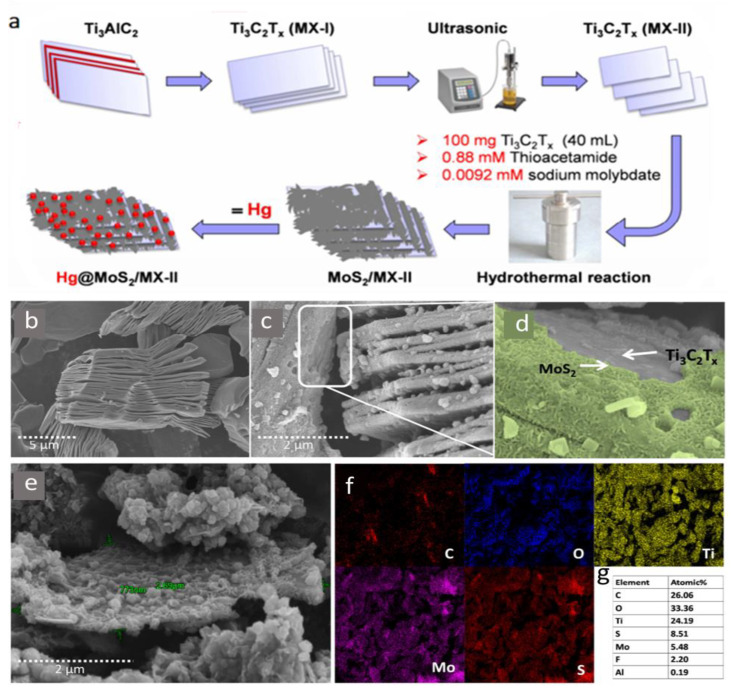
(**a**) Synthesis of MoS_2_/MX-II nanocomposite. SEM images of (**b**) Ti_3_C_2_T_x_, (**c**,**d**) MoS_2_/MX-I, and (**e**) MoS_2_/MX-II. (**f**) Elemental mapping of elements in MoS_2_/MX-II. (**g**) Elemental content (at %) of MoS_2_/MX-II determined by SEM-EDS. Adapted and reproduced with permission from [63]. Copyright 2020 Elsevier.

**Figure 12 nanomaterials-10-00885-f012:**
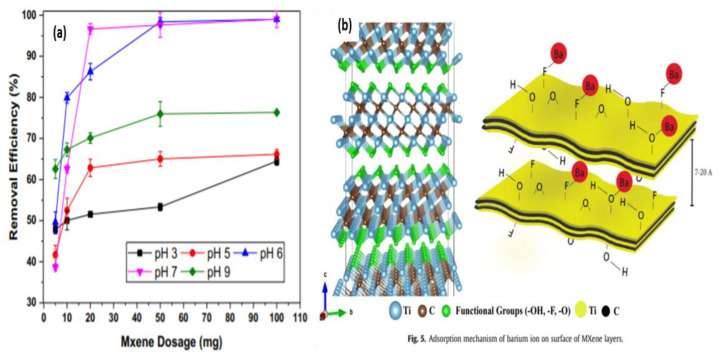
(**a**) Effect of pH on adsorption capacity and removal efficiency. (**b**) Adsorption mechanism of barium ion on the MXene surface. Adapted and reproduced with permission from [64]. Copyright 2017 Elsevier.

**Table 1 nanomaterials-10-00885-t001:** Some examples of MXene-based nanostructures for heavy metal removal reported elsewhere.

MXene	Preparation Method	Toxic Metals	Adsorption Conditions			Adsorption Capacity/Removal Efficiency	Mechanism	Ref.
			pH	Time	Temp			
Ti_3_C_2_(OH/ONa)*x*F_2−*x*_	Chemical etching of (Ti_3_AlC_2_) by HF followed by alkalization treatment with NaOH.	Pb(II)	5.8–6.2	2 min	323 K	140.1 mg/g	Adsorption	[48]
Ti_3_C_2_T*_x_*	-	Pb(II)	6	2 h	293 K	36.6 mg/g	Adsorption	[49]
Biosurfactant- functionalized Ti_2_CT_X_ MXene nanosheets	Chemical etching of (Ti_3_AlC_2_) by LiF+HCl solution followed by the addition of either CS, LS, or EHL solutions.	Pb(II)	5	24 h	30 °C	232.9 mg/g	Adsorption	[50]
(Ti_3_C_2_Tx- KH570)	Chemical etching of (Ti_3_AlC_2_) by 40% HF and then modified by KH570 with mechanical aggigtation to oxidize the MXene into TiO_2_ particles.	Pb(II)	1–6	2 h	30 °C	147.97 mg/g	Adsorption	[51]
Mxene/alginate	Adding MXene powder to varying amounts of sodium alginate.	Pb(II)	5–7	15 min	~50 °C	382.7 mg/g	Adsorption	[52]
Ti_3_C_2_T_x_	Chemical etching of (Ti_3_AlC_2_) by 10% HF followed by intercalation and ultrasonication	Cr(VI)	5	72 h	Room temperature	250 mg/g	Reduction/adsorption	[53]
Ti_3_C_2_	Chemical etching of (Ti_3_AlC_2_) by 40% HF followed by intercalation and sonication.	Cr(VI)	-	14 h	298 K	80 mg/g	Adsorption	[54]
TiO_2_-C (u-RTC)	Chemical etching of (Ti_3_AlC_2_) by LiF+HCl solution followed by in situ decomposition of the MXene in a mixed solution of ethylene glycol (EG), FeCl_3,_ and isopropyl alcohol (IPA).	Cr(VI)	3–6	120 min	-	~225 mg/g	Adsorption	[55]
Ti_3_C_2_/TiO_2_	Chemical etching of (Ti_3_AlC_2_) by HF followed by hydrothermal heating (160 °C) to obtain Ti_3_C_2_/TiO_2_ particles.	Cr(VI)	acidic	12 min	-	Reduction Efficiency 99.35%	Reduction/ adsorption	[56]
nZVI-alk-Ti_3_C_2_	Chemical etching of (Ti_3_AlC_2_) by HF followed by alkalization treatment with KOH and then adding NaBH4 to reduce Fe^2+^ into nZVI.	Cr(VI)	2	~1500 min	-	194.87 mg/g	Adsorption	[57]
Ti_3_C_2_T_x_/PmPD-5/1	Intercalation of poly(m-phenylenediamine) (PmPD) into regular MXene sheets.	Cr(VI)	2	~700 min	-	540.47 mg/g	Reduction/ adsorption	[58]
Ti_3_C_2_T_x_	Chemical etching of (Ti_3_AlC_2_) by HF followed by intercalation and ultrasonication	Cu(II)	5	3 min	298 K	78.45 mg/g	Reduction/ adsorption	[59]
Amino acids modified MXenes (Ti_3_C_2_T_X_-PDOPA)	Prepared through Single-step via self-polymerization of DOPA then adding it to the MXene.	Cu(II)	7	1 h	298 K	18.36 mg/g	Adsorption	[60]
Magnetic Ti_3_C_2_T_X_ nanocomposite	Magnetizing MXene flakes through the deposition of Fe_2_O_3_ and Fe_3_O_4_ nanoparticles on its surface.	Hg(II)	6	24 h	298K	1128.41 mg/g	Adsorption	[61]
Ti_3_C_2_T_X_ core-shell spheres containing sodium alginate	Chemical etching of (Ti_3_AlC_2_) by NH_4_F followed by ultrasonication under Ar gas with sodium alginate (SA) powder.	Hg (II)	4.5	24 h	298 K	932.84 mg/g	Adsorption	[62]
Molybdenum-disulfidefunctionalized MXenes (MoS_2_/MX)	Synthesized by a facile hydrothermal treatment method.	Hg (II)	2–11	2 min	-	1435.2 mg/g removal efficiency of 98.5%	Adsorption	[63]
Ti_3_C_2_T_x_	Chemical etching of (Ti_3_AlC_2_) by HF followed by intercalation and ultrasonication	Ba(II)	7	2 h	25 °C	9.3	Adsorption	[64]
Ti_3_C_2_(OH)_x_F_1−x_	Chemical etching of (Ti_3_AlC_2_) by followed by magnetic ferric oxide intercalation.	PO_4_^3−^	2.5–6	~250 min	-	2400 mg/g	Adsorption	[65]

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
