# Peer review of "Unveiling Fabrication and Environmental Remediation of MXene-Based Nanoarchitectures in Toxic Metals Removal from Wastewater: Strategy and Mechanism"

_nanomaterials, 2020, doi:10.3390/nano10050885_

Round 1
Reviewer 1 Report
This paper is quite outside my own field of expertise, but, as a review I found it both interesting and informative. I am not in a position to comment at length on the science, but the paper appears to me to be cogent, current and well argued. The figures are well presented and legends are appropriate. I therefore leave it to the editor to decide whether the manuscript is appropriate for the journal.
As a native english speaker, I could understand the authors' writing, but I am concerned that their use of florid and rather hyperbolic language is not suited to a scientific paper, and strongly recommend that the manuscript be reviewed, in depth, by someone from whom English is their primary language and who is expert in scientific writing, particularly the abstract.
Author Response
Reviewer 1
We would like to thank the reviewer 1 for his/her critical and insightful comments on the paper, which we think helped to improve the quality and clarity of this manuscript. We hope that our revisions and adaptations are adequate and reflect all the suggestions of reviewer 1. Our detailed responses to reviewers are given below.
Comments and Suggestions for Authors
This paper is quite outside my own field of expertise, but, as a review I found it both interesting and informative. I am not in a position to comment at length on the science, but the paper appears to me to be cogent, current and well argued. The figures are well presented and legends are appropriate. I therefore leave it to the editor to decide whether the manuscript is appropriate for the journal.
As a native english speaker, I could understand the authors' writing, but I am concerned that their use of florid and rather hyperbolic language is not suited to a scientific paper, and strongly recommend that the manuscript be reviewed, in depth, by someone from whom English is their primary language and who is expert in scientific writing, particularly the abstract.
Reply
Many thanks for your positive feedback. We have sent our manuscript to the MDPI for English editing, and they have revised it.
Reviewer 2 Report
This paper reviews the current literature on toxic metal removal using Mxene-based materials. The current manuscript has some drawbacks that need to be improved before publication.
- The "heavy metals" term has been recently not accepted by the scientific community and it is suggested to be replaced by “toxic metals”. The expression “heavy metals” was widely used by environmental scientists. However, it is encouraged to avoid using this term in a scientific publication. IUPAC reports specified that this expression had no coherent scientific basis. Please check the following reports for your consideration:
Pourret, O., Bollinger, J.-C., 2017. “Heavy metal” - What to do now: To use or not to use? Sci. Total Environ. 610–611, 419–420.
Chapman, P.M., 2007. Heavy metal - music, not science. Environ. Sci. Technol. 41, 6C.
Duffus, J.H., 2002. “Heavy metals” a meaningless term? (IUPAC Technical Report). Pure Appl. Chem. 74, 793–807.
- Part 2 heading should be changed to “Methods for toxic metal removal from wastewater”
- Part 4: The current literature on MXenes preparation should be summarized and compared through a table instead of reprinting the previous publication’s figures.
- An additional part, for example, “Limitations of MXenes application for toxic metal removal from wastewater” should be added.
- I disagreed on the re-print of too many previous publications’ figures even with the permission from their authors. Just some figures with some modifications to the original (about structure and theory) are recommended.
Author Response
We would like to thank the reviewer 2 for his/her critical and insightful comments on the paper, which we think helped to improve the quality and clarity of this manuscript. We hope that our revisions and adaptations are adequate and reflect all the suggestions of reviewer 2. Our detailed responses to reviewers are given below.
Comments and Suggestions for Authors
This paper reviews the current literature on toxic metal removal using Mxene-based materials. The current manuscript has some drawbacks that need to be improved before publication.
- The "heavy metals" term has been recently not accepted by the scientific community and it is suggested to be replaced by “toxic metals”. The expression “heavy metals” was widely used by environmental scientists. However, it is encouraged to avoid using this term in a scientific publication. IUPAC reports specified that this expression had no coherent scientific basis. Please check the following reports for your consideration:
Pourret, O., Bollinger, J.-C., 2017. “Heavy metal” - What to do now: To use or not to use? Sci. Total Environ. 610–611, 419–420.
Chapman, P.M., 2007. Heavy metal - music, not science. Environ. Sci. Technol. 41, 6C.
Duffus, J.H., 2002. “Heavy metals” a meaningless term? (IUPAC Technical Report). Pure Appl. Chem. 74, 793–807.
Reply
Many thanks for your positive feedback and helpful comment. We have replaced the "heavy metals" term by “toxic metals”. Kindly see the highlighted terms in the manuscript.
- Part 2 heading should be changed to “Methods for toxic metal removal from wastewater”
Reply
Many thanks for your positive feedback and helpful comment. We have changed the heading of part 2 to ‘’Methods for Toxic Metal Removal from Wastewater’’. Kindly see the highlighted texts in the manuscript.
- Part 4: The current literature on MXenes preparation should be summarized and compared through a table instead of reprinting the previous publication’s figures.
Reply
Many thanks for your positive feedback and helpful comment. In Table 1, we have summarized the preparation methods for MXenes. Also, Figure 1 summarizes the main methods for etching and delamination of MXenses. Kindly see Table 1 in the manuscript.
- An additional part, for example, “Limitations of MXenes application for toxic metal removal from wastewater” should be added.
Reply
Many thanks for your positive feedback and helpful comment. We have added an additional section entitled ‘’Limitations of MXenes application for toxic metal removal’’. Kindly see the highlighted section’’4.7.’’ in the manuscript.
- I disagreed on the re-print of too many previous publications’ figures even with the permission from their authors. Just some figures with some modifications to the original (about structure and theory) are recommended.
Reply
Many thanks for your positive feedback and helpful comment. We have removed 7 figures, but we kept only the most breakthrough results. Kindly see the manuscript.
Reviewer 3 Report
The article (Manuscript Number nanomaterials-772257) analyzes the most recent developments if the field of unveiling fabrications and environmental remediation of mxene-based nanoarchitectures in heavy metals removal from wastewater.
In my view, this review was carefully prepared and organized well. The overall originality of the review concept used here is high. Appealingly, it didn't read like a series of paper summaries, but like a narrative of the recent work in this area. To someone unfamiliar with the area (likely the majority of those who will read this article), I think the information is presented in quite a digestible format. The figures were chosen judiciously, and are generally of high quality.
Nevertheless, I would recommend publication of this review in Nanomaterials on the condition a minor revision of the manuscript will be carried out and the following points will be taken into consideration. I have only a few suggestions to improve this manuscript, which are listed below.
Detailed comments:
- The abstract needs to be well written with future prospects of the work and describe in short the concept of MXene-based nanoarchitectures in heavy metals removal from wastewater.
- More detailed advantagess of the present field must be mentioned in Introduction.
- The conclusion reflects an overall summary of the field with further extension and include future prospective - I would suggest clarifying this section.
- The chapter appears to be a collection of data from research papers, however, authors self-opinion is of importance while drafting a chapter of this type.
- I urge the authors to include the complete authorship of referenced works in the references section (if permitted under journal guidelines).
After completing the above-mentioned corrections this work will be more readable. Therefore, it will be useful for the readers of the Nanomaterials.
Author Response
We would like to thank the reviewer 3 for his/her critical and insightful comments on the paper, which we think helped to improve the quality and clarity of this manuscript. We hope that our revisions and adaptations are adequate and reflect all the suggestions of reviewer 3. Our detailed responses to reviewers are given below.
Comments and Suggestions for Authors
The article (Manuscript Number nanomaterials-772257) analyzes the most recent developments if the field of unveiling fabrications and environmental remediation of mxene-based nanoarchitectures in heavy metals removal from wastewater.
In my view, this review was carefully prepared and organized well. The overall originality of the review concept used here is high. Appealingly, it didn't read like a series of paper summaries, but like a narrative of the recent work in this area. To someone unfamiliar with the area (likely the majority of those who will read this article), I think the information is presented in quite a digestible format. The figures were chosen judiciously, and are generally of high quality.
Nevertheless, I would recommend publication of this review in Nanomaterials on the condition a minor revision of the manuscript will be carried out and the following points will be taken into consideration. I have only a few suggestions to improve this manuscript, which are listed below.
Detailed comments:
1. The abstract needs to be well written with future prospects of the work and describe in short the concept of MXene-based nanoarchitectures in heavy metals removal from wastewater.
Reply
Many thanks for your positive feedback and helpful comment. We have sent our manuscript to the MDPI for English editing, and they have revised it. We have also added the concept of MXene-based nanoarchitectures in heavy metals. Kindly see the highlighted sentences in the manuscript.
2. More detailed advantages of the present field must be mentioned in Introduction.
Reply
Many thanks for your positive feedback and helpful comment. In the Introduction, we have added the advantages of the adsorption method for the removal of heavy metals compared to other methods as well as the benefits of MXenes compared to other materials. Kindly see the highlighted sentences in the Introduction.
3. The conclusion reflects an overall summary of the field with further extension and include future prospective - I would suggest clarifying this section.
Reply
Many thanks for your positive feedback and helpful comment. We have added an additional section entitled’’ 4.7. Limitations of MXenes Application for Toxic Metal Removal’’, which discuss all the limitations and drawbacks of MXenes in heavy metal removal as well as future prospects for defeating their limitations. Kindly see the highlighted section 4.7. in the Introduction. On the other hand, the MDPI for English editing has revised the whole manuscript.
4. The chapter appears to be a collection of data from research papers, however, authors self-opinion is of importance while drafting a chapter of this type.
Reply
Many thanks for your positive feedback and helpful comment. In section 4.7 entitled’’ Limitations of MXenes Application for Toxic Metal Removal’’, we have provided our self-opinion on the limitations of MXenes rooting from preparation and properties to removal of heavy metal from wastewater. Kindly see the highlighted section 4.7. in the introduction.
5. I urge the authors to include the complete authorship of referenced works in the references section (if permitted under journal guidelines).
Reply
Many thanks for your positive feedback and helpful comment. We have added the complete authorship of referenced works and adjusted the reference style according to the journal guidelines.
After completing the above-mentioned corrections this work will be more readable. Therefore, it will be useful for the readers of the Nanomaterials.
Round 2
Reviewer 2 Report
The manuscript was significantly revised and can be now accepted for publication.